# Role of microRNAs in Chronic Lymphocytic Leukemia

**DOI:** 10.3390/ijms241512471

**Published:** 2023-08-05

**Authors:** Francesco Autore, Alice Ramassone, Luca Stirparo, Sara Pagotto, Alberto Fresa, Idanna Innocenti, Rosa Visone, Luca Laurenti

**Affiliations:** 1Dipartimento di Diagnostica per Immagini, Radioterapia Oncologica ed Ematologia, Fondazione Policlinico Universitario A. Gemelli IRCCS, 00168 Roma, Italyalberto.fresa@guest.policlinicogemelli.it (A.F.); idanna.innocenti@policlinicogemelli.it (I.I.); luca.laurenti@unicatt.it (L.L.); 2Center for Advanced Studies and Technology (CAST), G. d’Annunzio University, 66100 Chieti, Italy; alice.ramassone@unich.it (A.R.); sara.pagotto@unich.it (S.P.); r.visone@unich.it (R.V.); 3Department of Medical, Oral and Biotechnological Sciences, G. d’Annunzio University, 66100 Chieti, Italy; 4Sezione di Ematologia, Dipartimento di Scienze Radiologiche ed Ematologiche, Università Cattolica del Sacro Cuore, 00168 Roma, Italy

**Keywords:** CLL, microRNA, therapy

## Abstract

Chronic Lymphocytic Leukemia (CLL) is the most common form of leukemia in adults, with a highly variable clinical course. Improvement in the knowledge of the molecular pathways behind this disease has led to the development of increasingly specific therapies, such as BCR signaling inhibitors and BCL-2 inhibitors. In this context, the emerging role of microRNAs (miRNAs) in CLL pathophysiology and their possible application in therapy is worth noting. MiRNAs are one of the most important regulatory molecules of gene expression. In CLL, they can act both as oncogenes and tumor suppressor genes, and the deregulation of specific miRNAs has been associated with prognosis, progression, and drug resistance. In this review, we describe the role of the miRNAs that primarily impact the disease, and how these miRNAs could be used as therapeutic tools. Certainly, the use of miRNAs in clinical practice is still limited in CLL. Many issues still need to be solved, particularly regarding their biological and safety profile, even if several studies have suggested their efficacy on the disease, alone or in combination with other drugs.

## 1. Introduction

Chronic lymphocytic leukemia (CLL) is the most common form of leukemia in adults, with a higher prevalence in the elderly population [1]. It is characterized by the accumulation of mature B cells expressing CD5/CD19 surface markers. The clinical course of CLL varies significantly, yet the utilization of multiple biological markers aids in the clinical management of patients [2,3,4]. However, certain instances exhibit conflicting prognostic markers, necessitating the discovery of new parameters capable of correlating disease activity status with clinical outcome.

Molecular mechanisms that are deregulated are the B-cell Receptor (BCR) signaling, apoptosis, the Ataxia Telangiectasia Mutated (ATM)/Tumor Protein 53 (TP53) tumor suppressor pathway, autocrine cytokines, and interaction with microenvironment factors such as Interleukin-4 (IL-4), CD40 ligand, and accessory cells [5,6]. Several of these pathways have recently gained relevance as they can be inactivated by specific small molecules, which have shown clinical efficacy for the treatment of hematologic malignancies. Specifically, Ibrutinib, Idelalisib, and Fostamatinib enable BCR signaling, which has an essential role in the survival and proliferation of mature B lymphocytes and also affects the regulation of CLL B lymphocyte trafficking in lymphoid organs. Venetoclax is a B-cell Lymphoma 2 (BCL-2) inhibitor that promotes the apoptosis of cancer cells [2,4,7,8,9,10,11,12,13,14,15,16,17,18].

In onco-pathogenesis studies, the role of microRNAs (miRNAs or miRs) has been highlighted. MiRNAs are small endogenous single-stranded noncoding RNA molecules 20–22 nucleotides long, differentially expressed in different tumor types, whose deregulation may have both oncogenic and onco-suppressor functions; miRNAs regulate gene expression at the post-transcriptional levels by targeting the 3′UTR of messenger RNAs (mRNA), resulting in mRNA degradation and/or translation inhibition [5]. The generation of miRNAs is the result of a multistep process: initially, the miRNA is transcribed by the RNA Polymerase II/III, resulting in the formation of the primary miRNA (pri-miRNA); the pri-miRNA is processed in the pre-miRNA in the nucleus by Drosha, exported in the cytoplasm through the Exportin 5, and further processed by Dicer in the mature miRNA. In the cytoplasm, the miRNA is incorporated into the RNA-induced silencing complex (RISC), and after the binding of a target mRNA through the recognition of a complementary sequence at its 3′UTR, miRNA drives the mRNA downregulation [19]. Aberrant expression of miRNAs appears to be implicated in the onset of numerous diseases. Ever since the first demonstration of the association between miRNAs and cancer, it has been clear that these genes may play a role in the clinical management of cancer patients [20]. In CLL, miRNAs profiles can be used to distinguish normal B-cells from malignant CLL cells, and the deregulation of specific miRNAs has been associated with prognosis, progression, and drug resistance in CLL [21,22].

Our review aims to describe the close relationship between miRNAs and CLL, focusing on the miRNAs that most impact the disease, and how miRNAs could be used as therapeutic tools.

## 2. microRNAs in CLL

### 2.1. miR-15a/16-1

Back in 2002, Calin and colleagues identified two small non-coding RNAs, *miR-15a* and *miR-16-1*, within a 30 kb region at the 13q14 chromosomal region frequently lost in CLL patients. The deletion of *miR-15a/16-1* represents the first genetic alteration of a miRNA locus identified in human disease and occurs in around 70% of CLL patients [20]. The deletion correlates with *miR-15a* and *miR-16-1* downregulation [20,23] and associates with an indolent form of the disease [24]. Subsequently, a detailed high-resolution analysis showed that 13q14 deletion is heterogeneous in length, and *miR-15a/16-1* expression is lower in patients that have a larger 13q14 deletion than in patients with shorter or no 13q14 deletion. However, the authors also disclosed a few patients showing a low expression of *miR-15a* and *miR-16-1* (as those observed in CLL cases with larger 13q14 deletion), but there was no detectable alteration at 13q14 chromosomal region either by FISH or SNP array [25]. This finding could be explained, at least in part, by the existence of microdeletions of *miR-15a/16-1* locus: indeed, a recent analysis combining FISH and qPCR revealed that around 34% of CLL samples with no 13q14 deletion show a microdeletion of *miR-15a/16-1* locus [26].

Further studies highlighted that several mechanisms affect *miR-15a/16-1* expression and that their downregulation is not only due to 13q14 deletion. For instance, a germ-line mutation in *miR-15a/16-1* primary transcript was identified in two CLL patients who had monoallelic 13q14 deletion; the mutation, which consists of a C > T substitution 7 nucleotides downstream *miR-16-1* precursor, reduces miRNAs expression both in vitro and in vivo [27]. Similarly, a germ-line mutation in *miR-15a/16-1* primary transcript was also identified in New Zealand Black (NZB) mice, which spontaneously developed B-cell lymphoproliferative diseases. The mutation, consisting of a T > A substitution 6 nucleotides downstream *miR-16-1* precursor, decreases *miR-15a* and *miR-16-1* expression by impairing their processing [28,29]. Subsequently, Veronese and colleagues identified a single nucleotide polymorphism in two CLL patients (rs115069827) at about 100 nucleotides upstream *miR-15a* precursor, which abrogates the maturation of *miR-15a/16-1* primary transcript [30].

Beyond these, other factors have been identified as causing the aberrant miRNAs expression through the impairment of the miRNAs biogenesis pathway. Firstly, since *miR-15a/16-1* lies within the *Deleted In Lymphocytic Leukemia 2* (DLEU2) host gene, regulative factors affecting *DLEU2* expression also affect *miR-15a/16-1* expression. For instance, the oncoprotein MYC (MYC proto-oncogene) acts as a negative regulator of both *DLEU2* and *miR-15a/16-1* expression through the binding of two alternative *DLEU2* promoters [31]. At the same time, the B-cell-specific activator protein (BSAP) negatively regulates *DLEU2* and *miR-15a/16-1* by directly binding to the *DLEU2* promoter. In CLL cells the expression of BSAP is higher than in normal B cells, and its downregulation restores *miR-15a/16-1* expression in peripheral blood mononuclear cells from CLL patients [32]. Other factors involved in *miR-15a* and *miR-16-1* regulation are the histone deacetylases (HDACs), which are overexpressed in CLL and able to silence *miR-15a* and *miR-16-1* expression in about 35% of CLL patients; their inhibition restored the expression of both miRNAs [33]. Furthermore, the transcription mechanism itself was found to be deregulated: indeed, the locus of *miR-15a/16-1* shows an allele-specific transcription mechanism that involves both the canonical RNA Polymerase II (RPII) and the RNA Polymerase III (RPIII). Specifically, while the transcription of one allele is dependent on the *DLEU2* host gene and mediated by the RPII, the transcription of the other allele was independent of the *DLEU2* host gene and mediated by the RPIII. Interestingly, the authors discovered this particular transcription mechanism in CLL patients where the 13q14 deletion removed the allele transcribed by the RNA Polymerase II [30]. Finally, the maturation of *miR-15a* and *miR-16-1* was found to also be affected by RNA binding proteins in CLL. The adenosine deaminase RNA specific B1 (ADARB1) blocks *miR-15a* and *miR-16-1* maturation at the drosha ribonuclease III (DROSHA) processing level, but this mechanism is abrogated when the RNA-binding domain or the nuclear localization of ADARB1 was deleted [34].

In 2002, *miR-15a* and *miR-16-1* were found to be downregulated in about 70% of CLL patients [20], and it appeared that they could have a crucial role in CLL pathogenesis. A few years later, the same group demonstrated that both *miR-15a* and *miR-16-1* negatively regulated the antiapoptotic factor BCL-2 at the post-transcriptional level, and that their over-expression induced apoptosis in the leukemic cell line through BCL-2 downregulation [35]. Thereafter, combining experimental and bioinformatic approaches, the authors also identified several oncogenes modulated by *miR-15a/16-1*, such as BCL-2, Myeloid Cell Leukemia 1 (MCL-1), and Jun proto-oncogene (JUN), which directly or indirectly impair cell cycle and apoptosis [36]. Those data were further corroborated by another group, which highlighted that *miR-15a* and *miR-16-1* targeted MCL-1 in primary CLL cells [33]. The involvement of *miR-15a* and *miR-16-1* in cell cycle regulation was next confirmed by Klein and colleagues in murine models. The authors developed two different models, one lacking the minimal deleted region (MDR) at 13q14, and another one specifically lacking only *miR-15a/16-1* locus. They found that the deletion of *miR-15a/16-1* accelerated the proliferation of murine B cells by modulating their G_0_/G_1_-S phase transition, and this was exerted by the downregulation of several cyclins and cyclin-dependent kinases such as Cyclin D1 (CCDD1), Cyclin D2 (CCND2), Cyclin D3 (CCND3), Cyclin E1 (CCNE1), Cyclin-Dependent Kinase 4 (CDK4), Cyclin-Dependent Kinase 6 (CDK6), and Minichromosome Maintenance complex component 5 (MCM5) [37]. Recently, the onco-embryonic surface protein Receptor tyrosine kinase-like Orphan Receptor 1 (ROR1) has been identified as a novel target of *miR-15a/16-1*; this protein was found to be over-expressed in more than 90% of CLL patients, and its overexpression was associated with high levels of BCL-2 and low levels of *miR-15a/16-1* [38].

Note that *miR-15a* and *miR-16-1* do not only target oncogenes but also the tumor suppressor protein TP53. Indeed, a high level of *TP53* mRNA was found to be associated with reduced levels of *miR-15a* and *miR-16-1*, and an over-expression of BCL-2 [39]. Accordingly, *miR-15a* and *miR-16-1* directly targeted and downregulated the TP53 protein in primary CLL cells; however, at the same time, TP53 transactivated *miR-15a/16-1* in leukemic cells, indicating the existence of a regulatory feedback loop between *miR-15a/16-1* and TP53 [40].

#### Therapeutic Strategies

Several approaches have been suggested to take advantage of *miR-15a* and *miR-16-1* activity in CLL: two different groups demonstrated that in vivo delivery of *miR-15a* and *miR-16-1* was able to induce tumor regression in NOD/Shi-scid,γcnull (NSG) previously engrafted with CLL cells [41], and to reduce malignant B-1 cells in NZB mice [42]. Other studies highlighted that the modulation of factors that either target or are a target of *miR-15a/16-1* induces cell death. Kasar and colleagues demonstrated that combining both the HDACa inhibition and the BSAP knockdown, *miR-15a*, and *miR-16-1* expression, caused an increase in cell death [32]. Thereafter, Rassenti and colleagues demonstrated the therapeutic potential of blocking two *miR-15a/16-1* targets, ROR1 and BCL-2: the authors found that cirtuzumab, an anti-ROR1 monoclonal antibody, enhances the cytotoxic activity of venetoclax, which specifically targets BCL-2 [38].

### 2.2. miR-29 Family

The *miR-29* family consists of *miR-29a*, *miR-29b-1*, *miR-29b-2*, and *miR-29c*, generated from two primary transcripts: *pri-miR-29a/b1* cluster and *pri-miR-29b2/c* cluster, located at chromosomes 7q32.3 and 1q32.2, respectively. *MiR-29a* is the most represented and stable, due to its cytosine residue at nucleotide position 10. This family is known to be implicated in many diseases such as osteoarthritis, osteoporosis, cardiorenal disease, and immune disease [43,44].

The *MiR-29* family is particularly important for the regulation of the proliferation and differentiation of B and T lymphocytes. Repression of *miR-29*s expression via AKT and MYC pathways is associated with a loss of apoptosis and several B-cell malignancies, particularly lymphomas; the oncogenic transcription factor MYC was shown to regulate the expression of all *miR-29* family members by binding the promoter region of *miR-29b-1/a* and decreasing promoter activity by 50% [45].

In 2006, Pekarsky and colleagues found out that 75% of patients with aggressive CLL and deletion of 11q and 56% of patients with aggressive CLL without 11q deletion had a high expression of the oncogenic protein T-cell Leukemia/Lymphoma 1 (TCL-1), while 65% of patients with indolent CLL showed a low expression of this protein [46]. This was consistent with the evidence that high levels of TCL-1 expression correlate with unmutated Immunoglobulin heavy chain variable region (IGHV) status and Zeta Chain Of T Cell Receptor Associated Protein Kinase 70 (ZAP70) positivity [47]. In this study, the authors found that *miR-29b* and *miR-181b* are down-regulated in aggressive CLL with 11q deletion and that their expression inversely correlates with the levels of TCL-1 [46]. However, Santanam and colleagues showed that *miR-29a* and *miR-29b* expression was 4.5-fold higher in indolent CLL when compared with normal CD19+ B-cells [48]. Subsequently, in 2010, Pekarsky and colleagues generated transgenic mice over-expressing *miR-29* in B-cells and showed that the immunophenotypic profile of spleen lymphocytes from transgenic mice had increased populations of CD5+CD19+IgM+ B-cells, a characteristic of CLL. At the age of 12–24 months a markedly expanded CD5+ B-cell population was evident in the spleens of 85% of the transgenic mice, even if only 20% of these animals developed frank leukemia and died of this disease. This was more consistent with the development of an indolent form of CLL, which increased the percentage of leukemic cells with age (at the age of 20–26 months, on average, more than 65% of all B-cells were CD5+). This finding raised questions about the role of *miR-29* in the development of CLL and whether this miRNA acts as a tumor suppressor or an oncogene. The author concluded that the over-expression of *miR-29* is not sufficient for the development of an aggressive CLL, while it may act as a trigger that initiates or at least significantly contributes to the pathogenesis of indolent CLL. In contrast, *miR-29* down-regulation contributes to the up-regulation of TCL-1 and the development of aggressive CLL [49]. MCL-1 is another important target of *miR-29b*: indeed, the enhanced expression of *miR-29b* reduced MCL-1 protein levels and facilitated apoptosis [50].

Recently, Sharma et al., by studying the intra-clonal CLL cell subpopulations that egressed the lymph nodes (CXCR4^dim^ CD5^bright^ cells), noted the *miR-29* family as downregulated in CXCR4^dim^ CD5^bright^ cells compared to CXCR4^bright^ CD5^dim^. In the same analysis, the tumor necrosis factor receptor-associated factor 4 (TRAF4) was around 2.4-fold upregulated. Moreover, patients with high levels of TRAF4 had significantly shorter survival (HR 2.4) and more aggressive disease [51]. TRAF was identified as a novel target of *miR-29*. TRAF4 overexpression enhances B-cell responsiveness to CD40 ligation, increasing the B-T cells interaction and, together with BCR activation, may be particularly important for the development of CLL phenotype [52]. Indeed, CXCR4^dim^ CD5^bright^ cells, with more activated BCR signaling, have higher levels of MYC protein, which downregulates *miR-29* by binding their promoter region [45], and, in turn, increases TRAF4 and CXCR4^bright^ CD5^dim^ cells. Finally, the authors observed that BCR activity in CLL represses *miR-29* and its inhibition by therapeutic agents, such as Ibrutinib or Idelalisib, increases levels of *miR-29* and consequently decreases levels of TRAF4 [51].

#### Therapeutic Strategies

As evidence of the therapeutic efficacy of *miR-29* in CLL, in 2019 Chiang and colleagues developed an immuno-nanoparticle-based *miR-29b* delivery formulation with selectivity to CLL cells but not normal B cells thanks to its target to ROR1, which is expressed in 95% of CLL cells but not in normal B cells [53]. Treatment with this agent increased the levels of intracellular *miR-29b* by around 600-fold and led to the downregulation of the DNA methyltransferase 1 (DNMT1), DNA methyltransferase 3 alpha (DNMT3A), and Sp1 transcription factor (SP1) in cancer cells, thus reducing CLL’s selective hypermethylation and restoring mechanisms of apoptosis. Moreover, treatment affects CLL cells but not normal B cells, therefore opening a way to clinical trials involving agents such as *miR-29b* in the setting of CLL or other tumors resistant to DNA hypomethylating agents [54].

### 2.3. miR-34 Family

The *miR-34* family is composed of three members: *miR-34a*, located at chromosome 1p36, and *miR-34b/c*, located within the 11q23 region [55,56], whose deletion represents one of the most common chromosomal abnormalities in CLL and is associated with a poor clinical outcome [24]. While *miR-34a* has its own primary transcript, *miR-34b/c* is located within the same primary transcript, at about 500 bp distance [57]. A deepest analysis of the 11q region in 178 CLL patients showed that *miR-34b/c* cluster localizes within a minimal deleted region, spanning close to the ATM gene [58]. A low level of *miR-34a* was associated with shorter treatment-free survival [59], poor response to Fludarabine-based chemotherapy [60,61,62], and can predict the development of Richter’ syndrome, a transformation of CLL in an aggressive lymphoma [63].

TP53 was discovered to directly activate *miR-34* family members. The overexpression of TP53 increases both the primary transcripts of *miR-34* and *miR-34* expression, while TP53 silencing reduced *miR-34a* levels; accordingly, the putative promoter region of both *pri-miR-34a* and *pri-miR-34b/c* shows a significative sequence conservation [57]. The TP53 network is a central player in regulating cell survival and DNA repair [64], and several authors showed that *miR-34*s induction recapitulates TP53 function, such as proliferation and apoptosis regulation: for instance, *miR-34a* directly affects CDK4, CDK6, CCND1, CCNE2, and MET proto-oncogene, receptor tyrosine kinase (MET) (Reviewed by He and colleagues [65]). In Chronic Lymphocytic Leukemia, TP53 is frequently deleted or mutated. TP53 gene maps within a 17p region frequently lost in CLL patients, and TP53 mutations occur in about 8% of CLL patients; both 17p deletion and TP53 mutation are associated with a poor outcome, and patients with TP53 mutations show resistance to Fludarabine-based chemotherapy [24,66,67]. The impact of TP53 regulation on *miR-34*s in CLL patients was subsequently demonstrated by different authors, which highlighted that *miR-34a* is downregulated in CLL patients carrying 17p/TP53 deletion, while *miR-34b/c* was undetectable [60,61,68,69]. As the TP53 mutation, the downregulation of *miR-34a* was also linked to Fludarabine resistance in CLL [60,61]. Regarding a possible mechanism explaining the role of *miR-34a* in refractory CLL, Zenz and colleagues demonstrated that this miRNA altered both the resistance to apoptosis and the DNA Damage Response (DDR), even without 17p deletion or TP53 mutation [60]. The involvement of *miR-34a* in DNA damage response was recently confirmed by Cerna and colleagues. The authors found that *miR-34a* is usually upregulated during DDR in CLL cells during FCR (Fludarabine, Cyclophosphamide, Rituximab) therapy; in this process, *miR-34a* downregulates the transcription factor Forkhead box P1 (FOXP1), thus limiting its ability to stimulate BCR signaling; the authors also demonstrated that low levels of *miR-34a* could be used as a biomarker of poor response in CLL patients treated with FCR [62].

Several factors affecting TP53 activity also impair *miR-34*s expression. Asslaber and colleagues demonstrated that a single nucleotide polymorphism within the intronic promoter of the MDM2 proto-oncogene (MDM2), a negative regulator of TP53 [70], named SNP309, affects *miR-34a* expression: indeed, by comparing *miR-34a* expression in patients carrying the GG genotype versus patients carrying the TT genotype, they found a significantly lower expression of *miR-34a* in the former group [59]. The GG genotype of the SNP309 in the MDM2 gene increases the expression of MDM2 attenuating the TP53 pathway [71], and was associated both with reduced overall survival and reduced treatment-free survival [72]; accordingly, low *miR-34a* levels were associated with shorter treatment-free survival and its overexpression in CLL cells induced apoptosis [59]. The activity of TP53 was found to be regulated by *miR-15a/16-1*: the overexpression of these miRs reduces both TP53 and *miR-34a*, *miR-34b*, and *miR-34c* levels; moreover, their overexpression increases ZAP70 level, a target of the *miR-34b/c* cluster [40].

Despite the presence of TP53, the methylation level of *miR-34b/c* promoter also plays an important role in controlling *miR-34b/c* expression; indeed, their promoter was found to be completely methylated in four CLL cell lines, while it was unmethylated in normal samples. The methylation of the *miR-34b/c* promoter leads to *miR-34b/c* downregulation and, as a consequence, to TP53 pathway alteration [73]. Accordingly, Deneberg and colleagues demonstrated that the *miR-34b/c* promoter was hypermethylated in about 48% of CLL patients and that the expression of *miR-34b/c* was inversely correlated to the DNA methylation levels [74].

#### Therapeutic Strategies

The *miR-34a* acts as a key regulator of tumor suppression in several tumor types by controlling different proteins involved in apoptosis, cell cycle regulation, differentiation, and chemoresistance [75]. In 2013, the first Phase 1 study consisting of miRNA-based cancer therapy was performed using a *miR-34a* liposomal mimic, MRX34; the clinical study (ClinicalTrials.gov identifier NCT01829971) approached various solid tumors and hematologic malignancies, demonstrating a dose-dependent modulation of relevant target genes in solid tumors. However, the study was prematurely terminated after the emergency of severe side effects and the death of four patients [76,77,78].

### 2.4. miR-17-92 Cluster

Back in 2004, Ota and colleagues identified a novel gene, designated “Chromosome 13 open reading frame 25 (C13orf25)”, which was overexpressed in B-cell lymphoma cell lines and diffuse large B-cell lymphoma patients with 13q31-q32 amplifications in cells from 70 patients [79]. Currently, this is known as the *miR-17-92 host gene (MIR17HG)* and contains the primary transcript of *miR-17-92 (pri-miR-17-92)* that is processed into seven different mature miRNAs: *miR-17*, *miR-18a*, *miR-19a*, *miR-19b-1*, *miR-20a*, *miR-20b*, and *miR-92a-1*. Gene duplications and deletions result in two *miR-17-92* paralogs: the *miR-106b-25* cluster on chromosome 7 and the *miR-106a-363* cluster on chromosome X, that bring the number of miRNAs present in the *miR-17-92* cluster to 15 overall. Chocholska and colleagues [80] highlighted the heterogenenous expression of the *miR-17-92* cluster members in CLL patients. They found a significantly higher *miR-17-5p* expression level in the CLL group with a high risk of progression (stage III/IV), 11q22.3 deletion, trisomy 12 and/or 17p13.1 deletions, expression of CD38 and ZAP70, when compared to the low-risk group. The expression of *miR-17-5p* at the time of diagnosis was higher in patients requiring therapy; regarding treatment response, patients with the progressive disease showed higher levels of *miR-17-5p* as compared to patients with the partial or complete response. By contrast, the presence of isolated del(13q14), a marker of better disease prognosis, was associated with a significantly lower *miR-17-5p* expression and higher *miR-19a-3p*, *miR-92a-1-5p* and *miR-20a-5p* expression compared to patients carrying unfavorable genetic aberrations. The higher expression of these miRNAs also correlated with negative ZAP70 and CD38 expression, while low *miR-20a* expression (<1.629) was strongly associated with shorter time to treatment (TTT) and PD. In multivariate analysis, low *miR-20a* expression remained an independent marker predicting short TTT for CLL patients, replying to the results obtained by Selven and colleagues [81] in colorectal cancer and opening a way to its use as a potential blood biomarker.

The expression of this cluster is regulated by many transcription factors, such as MYC, Myc-associated factor X-interacting protein 1 (MXI), B-cell lymphoma 3 (BCL-3), TP53, hypoxia-inducible factor-1 (HIF-1), and many others that act on binding its promoter to the 5′ untranslated region (5′-UTR) of *MIR17HG* [82]. In CLL, patients without TP53 expression and aggressive disease showed reduced *miR-17* and *miR-20* expression and increased *miR-19a/b* and *miR-92a* expression, whereas miR-18 levels remained unchanged. By contrast, patients who expressed TP53 wild type exhibited increased levels of *miR-17* and *miR-20*, unchanged levels of *miR-18*, *miR-19a/b*, and lower levels of *miR-92a*, suggesting that TP53 inactivation triggers the imbalanced expression of individual *miR-17-92* members, leading to overexpression of oncogenic miRNAs [83].

In 2012, Bomben and colleagues demonstrated that the microenvironment can improve the expression of *miR-17-92* cluster in CLL cells [84]. Unmutated IGHV CLL has a greater capacity to signal through the BCR upon antigen stimulation; accordingly, stimulation of CLL cells using the immunostimulatory cytosine–phosphate–guanosine (CpG), an agonist of the Toll-Like Receptor 9 (TLR9), increased the expression of 21 miRNAs in IGHV unmutated cells. Between these upregulated miRNAs, the ones belonging to the *miR-17-92* cluster or its paralogs (*miR-17*, *miR-20a*, *miR-20b*, *miR-17*, *miR-18a*, *miR-19b-1*, and *miR-92a-1*) were the most important regulators of the gene expression profile induced by CpG stimulation in IGHV-unmuted CLL cells. These miRNAs induced downregulation of genes such as the Zinc finger and BTB domain containing 4 (ZBTB4) and the Tumor Protein p53 Inducible Nuclear Protein 1 (TP53INP1), which regulate apoptosis through the Cyclin-Dependent Kinase inhibitor 1A (CDKN1A) and TP53 [85], and activated MYC pathways, strongly suggesting an associative interaction between MYC and *miR-17-92* in CLL.

#### Therapeutic Strategies

As already proposed by a few studies [86,87], oncogenic miRNAs (oncomiR) may be silenced by small oligonucleotides known as antagomiR, even if they have not been yet tested in clinical trials in this context. For this purpose, Dereani and colleagues designed a specific oligonucleotide targeting endogenous *miR-17*, known to be one of the most active in the *miR-17-92* cluster, named antagomiR17. The authors transfected MEC-1 CLL-like cells with antagomiR17 or scrambled controls and injected it in severe combined immunodeficiency (SCID) mice. In vivo tumors generated by MEC-1 cells in SCID mice showed a dramatic reduction in mass growth after treatment with antagomiR17, with its complete regression in one-fifth of cases (20%). The median overall survival (OS) of mice treated with antagomiR17 was significantly longer than that of mice treated with scrambled control or saline solution, and none of the mice showed significant toxicity. These data might open a new therapeutic strategy for those CLL patients who are still refractory to new therapies [88].

### 2.5. miR-155

*MiR-155* is a typical multifunctional miRNA associated with physiological and pathological processes, including immune response, inflammatory reaction, hypoxia, hematopoiesis, and tumorigenesis [89,90,91,92,93]. Indeed, *miR-155* is a well-established tumor-promoting miRNA, acting predominantly as an oncomiR, and it is one of the most commonly up-regulated miRNAs in several types of hematological malignancies (i.e., Burkitt Lymphoma, Hodgkin’s Lymphoma, some types of Non-Hodgkin’s Lymphoma, Acute Myeloid Leukemia, and CLL) [94,95,96,97,98], and solid tumors (i.e., breast, colon, cervical, pancreatic, lung, and thyroid cancer) [99,100,101,102,103].

*MiR-155* originates and is processed from the evolutionally conserved region of the host gene (*MIR155HG*), originally identified as the B-cell Integration Cluster (BIC) gene [104]. The BIC gene region localizes within the 21q21.3 on chromosome 21q21, a common retroviral integration site activated by viral promoter insertion in avian leukosis virus-induced B-cell lymphomas in chickens. In 2001, Tam W. identified a BIC homologous gene in humans and mice, with more than 70% of homology over 138 nucleotides among mice, humans, and chickens in exon III of human–BIC and mouse–BIC, and exon II of chicken-BIC [105]. However, BIC cDNAs isolated from these three species lacked a long open reading frame (ORF); moreover, none of the short ORFs present in the three cDNAs were conserved or showed significant homology to each other, making it very unlikely that these ORFs encoded functional polypeptides [105,106]. Further examination of the secondary structure of BIC RNA revealed the formation of an imperfect RNA duplex within the region of sequence homology in humans, mice, and chickens, suggesting that BIC might function as a non-protein-coding RNA. According to all these evidences, the original BIC gene has now been designated as the *MIR155* host gene that spans a 13 kb region that generates the BIC transcript, a 1500 bp noncoding *primary-miRNA-155* transcript (pri-miRNA) in exon III, which is further processed to the mature *miR-155-5p* and *miR-155-3p*^*(*)*^ strands that each contribute differently to post-transcriptional regulation of their target genes. The *miR-155-5p* is the functionally dominant and more abundant form (from 20-fold to 200-fold higher than *miR-155-3p*), however, despite this disparity in expression level, *miR-155-3p* possesses functional biological activity implicated in immune response and cancer [107,108,109,110].

*MiR-155* was identified as one of the most upregulated miRNAs in several solid and haematological malignancies. Additionally, the promoter region of *miR-155* contains different binding sites for transcription factors that contribute to regulating *miR-155* expression, such as SMAD family member 4 gene (SMAD4, -600 bp), the interferon-sensitive response element (ISRE, -311 bp), the interferon regulatory factors (IRF, -200 bp) [111,112], the AP-1, which is critical for B-cell activation [113], the Ets critical for *miR-155* induction by LPS [114], the hypoxia-inducible factor-1 alpha (HIF1-α) binding sites [115], and the nuclear factor-kappa B (NF-kB), which are essential for the Epstein–Barr virus latent membrane protein 1 (LMP1)-dependent activation involving NF-kB (by facilitating the binding of p65 to the *miR-155* promoter) [116]. *MiR-155* is one of the best-characterized oncomiRs in hematological malignancies. *BIC/miR-155* levels in humans are low in normal lymphoid tissues but accumulate in human B-cell lymphomas, Hodgkin lymphomas, and some subtypes of Non-Hodgkin’s lymphoma as the Diffuse large B-cell lymphoma, certain types of Burkitt lymphomas, acute myeloid leukemia, and CLL [94,97,98,117,118]. The concluding role of *miR-155* in the molecular mechanisms of B-cell development and lymphomagenesis was obtained in 2006 by the generation of a transgenic mouse overexpressing *miR-155* specifically in the B cells. These transgenic mice expressed *miR-155* under the control of the Eμ enhancer region (Eμ-miR-155) and developed an initially polyclonal expansion of pre-leukemic B cells proliferation, evident in the spleen and bone marrow, followed by high-grade B-cell Lymphoma resembling human disease approximately at the age of 6 months [119]. These findings show that *miR-155* is able to induce polyclonal expansion, supporting secondary genetic changes for a full transformation, and suggests a direct involvement of the *miR-155* in the initiation and/or progression of these diseases. In 2011, Vargova and colleagues demonstrated that v-myb myeloblastosis viral oncogene homolog (MYB) was overexpressed in a subset of B-CLL. In this report, they identified MYB binding sites onto the *MIR155HG* promoter near the TSS in primary B-CLL, resulting in the dysregulation of *miR-155*’s epigenetic status and its aberrantly elevated levels in CLL [120]. Two years later, in 2013, *miR-155* overexpression was found to be associated with B cells from individuals with monoclonal B-cell lymphocytosis (MBL), and even more in B cells from patients with CLL when compared with normal B cells from healthy individuals. Furthermore, it was demonstrated that *miR-155* expression levels in plasma samples collected before treatment were lower in CLL patients who achieved complete remission than in all others. Collectively, these data suggested that *miR-155* is a useful marker to identify cases of MBL that may progress to overt CLL and patients with CLL who may not respond well to therapy [118]. In 2014, *miR-155* was further associated with aggressive CLL [121]. In this study, it was determined that CLL with a high level of *miR-155* expressed lower levels of Src homology-2 domain-containing inositol 5-phosphatase 1 (SHIP1), a phosphatase that may suppress surface immunoglobulin, enhancing the sensitivity to BCR ligation compared to CLL with low levels of miR-155. Additionally, SHIP1 is a direct target of *miR-155* expression which, in turn, is positively regulated by crosstalk within the lymphoid tissue microenvironment, such as CD154 or the B-cell activating factor (BAFF), enhancing the BCR signaling, promoting proliferation in cancer cells, and potentially contributing to its association with adverse clinical outcomes in patients with CLL. Moreover, similar effects were also found in normal B cells stimulated via the expression of members of the tumor necrosis factor family of proteins (CD40 ligation with CD154), indicating a possible physiologic role of *miR-155* in regulating the B-cell response to BCR ligation [121]. Furthermore, *miR-155* was linked with aneuploidy and early cancer cell transformation, and it has been ascertained that *miR-155* overexpression directly affects the recruitment of three essential proteins to the kinetochores (BUB1, CENP-F, and ZW10), triggering chromosome alignment defects at the metaphase plate and increasing the rate of aneuploidy. On the other hand, during the advanced passages of cellular transformation, the RNA-binding protein heterogeneous nuclear ribonucleoprotein L (HNRNPL) binds to the polymorphic marker D2S1888 at the 3′UTR of the BUB1 mitotic checkpoint serine/threonine kinase (BUB1) gene, hampers *miR-155* targeting, and allows for the expansion and stabilization of most suitable clones in CLL [122,123]. Given these results, it is to be emphasized that *miR-155* has a deep impact on CLL development and progression.

#### Therapeutic Strategies

*MiR-155* overexpression has oncogenic activity in the majority of tumors, including CLL, therefore making it an interesting therapeutic target for the treatment of cancer. The first phase I clinical trial with a synthetic locked nucleic acid (LNA anti-miR) of *miR-155* inhibitor, Cobomarsen (MRG-106), started on February 2016 in patients with cutaneous T cell lymphoma (CTCL), CLL, DLBCL, and adult T cell leukemia/lymphoma (ATLL) (ClinicalTrials.gov; Identifier: NTC02580552). The first preliminary results obtained from the evaluation of six CTCL patients suggest that the cobomarsen is well-tolerated and results in therapeutic improvements [124]. In 2021, MRG-106 was tested in DLBCL cell lines, in corresponding xenograft mouse models, and a patient with aggressive ABC-DLBCL; the study demonstrated that Cobomarsen decreases cell proliferation in vitro and tumor volume in vivo, and that the compound reduced and stabilized tumor growth without any toxic effects in the patient [125]. However, the clinical trial is ongoing, and only future evaluations in patients with other cancers will yield more information on the clinical applicability of anti-*miR-155* therapy.

### 2.6. miR-181 Family

*MiR-181* family consists of four members: *miR-181a*, *miR-181b*, *miR-181c*, and *miR-181d*. They are localised in three different genomic clusters lying within three separate chromosomes: the *miR-181a-1/b-1* cluster is located on chromosome 1, the *miR-181a-2/b-2* cluster is placed on chromosome 9 and, finally, the *miR-181c/d* cluster is placed on chromosome 19 [126].

In CLL, the expression of *miR-181b* was found to vary according to disease stages: indeed, *miR-181b* is down-regulated in CLL compared to control samples [127,128,129,130], and its expression decreases during CLL progression, suggesting its evaluation as an important tool for monitoring the course of CLL [127]. Similarly to *miR-181b*, *miR-181a* appears to be downregulated during CLL progression, even though it was expressed at lower levels [128]. Visone and colleagues demonstrated that a reduction of *miR-181b* greater than 50% between sequential CLL samples and/or a *miR-181b* value lower than 0.005 at the starting time point was able to differentiate patients with stable disease from patients with progressive disease, and was associated with an increased risk to start treatment [127]. Thereafter, the authors demonstrated that *miR-181b* predicts the risk of progressive disease similarly to other markers such as ZAP70 and IGHV [128]. Accordingly, *miR-181b* results in being downregulated in aggressive cases of CLL with 11q deletion [131], and its low expression was linked to progression and cell death resistance in CLL [132].

*MiR-181b* plays an important role in the regulation of apoptosis: indeed, it downregulates both MCL-1, TCL-1, and BCL-2 [46,47,127,133,134,135,136,137,138,139]. *MiR-181b* significantly decreased the TCL-1 protein, and appeared to be the strongest TCL-1 inhibitor among *miR-181* family members; however, either a reverse or a direct correlation was found between *miR-181b* expression and TCL-1 protein expression in CLL, suggesting that multiple mechanisms orchestrate the miRNAs-TCL-1 interaction [46,140]. In vivo studies demonstrated the capability of *miR-181b* to reduce leukemic cell expansion, affecting proliferative, survival, and apoptotic pathways [140]; however, the usage of siRNA against TCL-1 did not achieve a high level of apoptosis, suggesting that other relevant targets different from TCL-1 exist and are important in mediating biological effects [140]. *MiR-181b* plays a role also in immune response regulation: recently, Di Marco and colleagues demonstrated that *miR-181b* modulates T cells’ cytotoxic activity against CLL cells: firstly, the author found that the expression of *miR-181b* increases in CLL cells by CD40-CD40L interaction. They demonstrated that the overexpression of *miR-181b* following the CD40 stimulation enhances the maturation of cytotoxic T lymphocytes and the death of CLL cells through the depletion of the anti-inflammatory cytokine interleukin-10 [141].

#### Therapeutic Strategies

As a natural TCL-1 inhibitor, *miR-181b* has been considered a drug candidate to treat CLL which over-expresses TCL-1 [133]. The Eµ-TCL-1 transgenic mouse model is characterized by the development of leukemia whose features recapitulate the aggressive human CLL. Bresin and colleagues demonstrated that the overexpression of *miR-181b* in murine splenocytes from Eµ-TCL-1 mice induces apoptosis; moreover, the in vivo administration of *miR-181b* reduced leukemia expansion and increased survival [140]. Zhu and colleagues also identified a synergistic activity of *miR-181b* with fludarabine, highlighting the ability of miRNAs to improve the efficacy of fludarabine to induce apoptosis [132].

## 3. Conclusions

The use of miRNA in clinical practice is still limited, especially in CLL. Many challenges have to be overcome to prompt the application of miRNAs in clinical routines, such as improving their specific cellular uptake by CLL cells and reducing the side effects on healthy cells. However, miRNAs are potent regulators affecting several cellular pathways in CLL, such as immunomodulation, proliferation, and cell death (Figure 1). Alone or combined with other drugs (Table 1), their use as therapeutic tools could improve the life of patients.

## Figures and Tables

**Figure 1 ijms-24-12471-f001:**
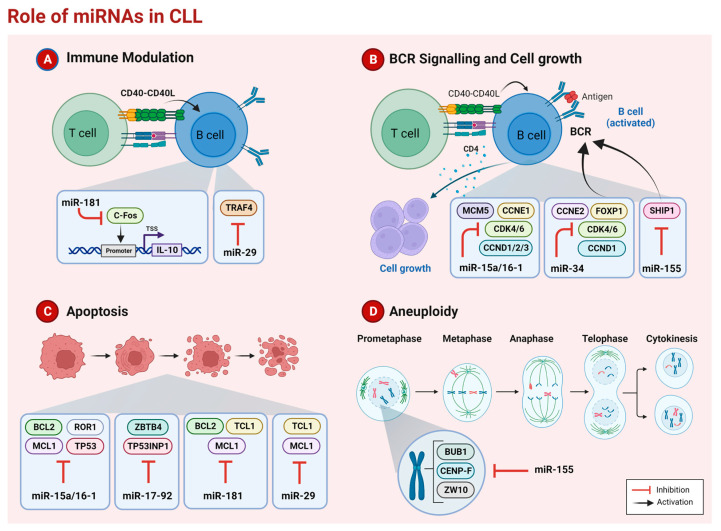
Schematic representation of mechanisms deregulated by microRNAs in Chronic Lymphocytic Leukemia (Created with Biorender.Com).

**Table 1 ijms-24-12471-t001:** MiRNA deregulated in CLL.

Family Name	Members	Targets		Therapeutic Strategies
*miR-15a/16-1*	*miR-15a* *miR-16-1*	All members	BCL-2 [35] MCL-1, JUN [36] ROR-1 [38] TP53 [39]	In vivo delivery of *miR-15a* and *miR-16-1* (preclinical) [41,42]
*miR-29*	*miR-29a* *miR-29b-1* *miR-29b-2* *miR-29c*	*miR-29b*	TCL-1 [46] MCL-1 [50] TRAF4 [52]	Immuno-nanoparticle-based *miR-29b* (preclinical) [54]
*miR-34*	*miR-34a* *miR-34b* *miR-34c*	All members	TP53 [57]	*miR-34a* liposomal mimic, MRX34. (phase 1 study, prematurely terminated after the emergency of severe side-effects) [76]
		*miR-34a*	CDK4 CDK6 CCND1 CCNE2 MET FOXP1 [65]	
		*miR-34b/c*	ZAP70 [40]	
*miR-17-92 cluster*	*miR-17* *miR-18a* *miR-19a* *miR-19b-1* *miR-20a* *miR-20b* *miR-92a-1*	*miR-17* *miR-19*	PTEN [82]	AntagomiR17 (preclinical) [88]
		*miR-92a-1* *miR-19*	Bim [82]	
		*miR-17* *miR-20a* *miR-20b* *miR-17* *miR-18a* *miR-19b-1* *miR-92a-1*	ZBTB4 TP53INP1 MYC [84]	
*miR-155*	*miR-155*		SHIP1 [121] BUB1 CENP-F ZW10 [122]	Cobomarsen, MRG-106 (phase 1 clinical trial) [125]
*miR-181*	*miR-181a-1* *miR-181a-2* *miR-181b-1* *miR-181b-2* *miR-181c* *miR-181d*	*miR-181b*	TCL-1 MCL-1 BCL-2 [127] c-FOS [141]	In vivo administration of *miR-181b* (preclinical) [140]

## Data Availability

Not applicable.

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
