# Peer review of "Role of microRNAs in Chronic Lymphocytic Leukemia"

_ijms, 2023, doi:10.3390/ijms241512471_

Round 1

Reviewer 1 Report

Autore et al provides a cohesive and succint review of some of the microRNAs (miRNA) reported to be involved in chronic lympothocytic leukemia (CLL). The format of the manuscript in which the authors first provide a description of the miRNA family and their relevance to CLL followed by their therapeutics potentials makes it easy for the readers to follow the story. Although the role of miRNAs in CLL is well-documented, an up-to-date account of the key miRNAs could be useful to the people working in the field. The following issues should be addresed prior to warranting publication:

Major issues:

1. It would be nice to summarize the miRNAs and their relevance to CLL in a table consisting of 1) the name of miRNA family; 2) the family members; 3) potential targets of miRNAs or relevance to CLL and 4) therapeutic strategies.

2. Lines 48-54 should be revised to cover miRNA biogenesis and mode of actions to introduce the concept to those who might not be familiar with it.

3. There are just too many grammar and typing errors that require extensive proofreading by a native speaker. Some parts of the manuscript are rather cohesive and well-written whereas some other parts are not.

4. The authors should cite more current studies.

5. There are inconsistencies in the use of terms. For example, the authors use “microRNA”, “miRNA” and “miR” interchangably. Similarty, “Author et al”, “Author et colleagues”, or “Author and colleagues”, giving the impression that different parts of the manuscript has been written by different authors without going over the last version. Please revise accordingly.

6. I suggest “Role of miRNAs in ........” as the title.

7. There are several paragraphs consisting of single sentences. Please combine such paragrapsh with the preceeding or following paragraphs. Please avoid paragraphs of single sentences.

Minor issues:

·         Line 55, “microRNAs” should be “miRNA”

·         Line 64, “a 30 kb at” should be “a 30 kb region at”; also, “loss” should be “lost”

·         Line 65, “represent” should be “represents”

·         Line 81, “consists in” should be “consists of”—there are several such errors throughout the text.

·         Line 89, “abrogate” should be “abrogates”

·         Line 139, “does” should be “do”; lines 142-143, “downregulate” and transactivate” should be past tense.

·         Line 171, “find” should be “found”

·         Line 191, “over expression” à overexpression

·         Line 198, “egpressed” should be “expressed”

·         Line 203, “ifentifie” should be “identified”

·         Line 226, “which” should be “whose”

·         Line 228, “an own” should be “its own”

·         Line 235, “transcrib” is a bit awkward. Maybe “activate”?

·         Line 284 act à acts

·         Line 290 were à was

·         Line 299, results à result

·         Lines 305-307- This sentence is awkward

·         Line 320, 50-UTR à 5’UTR

·         Line 328, “Unmuted ...have” à Unmutated..... has... Please change all “unmuted” to “unmutated”

·         Lines 438-439, hamper, allow à hampers, allows

·         Line 481, orcheastrates à orchestrate; line 486, are à is; line 488, play à plays

·         “In vitro” and “in vivo” should be italicized.

·         Many more typing and grammar errors......

There are just too many grammar and typing errors and awkward sentences. I believe that an extensive editing would be needed to warrant publication.

Author Response

Dear Editor,

Thank you for giving us the opportunity to revise and improve our manuscript “Role of microRNAs in Chronic Lymphocytic Leukemia”. We also want to thank the reviewers for their constructive suggestions.

The main text has been edited to fix spelling errors and improve readability following the reviewers’ comments. The point-by-point replies to the major reviewers’ comments are below.

Changes in the manuscript are in red.

Reviewer#1

  1. It would be nice to summarize the miRNAs and their relevance to CLL in a table consisting of 1) the name of miRNA family; 2) the family members; 3) potential targets of miRNAs or relevance to CLL and 4) therapeutic strategies.

As suggested by the reviewer, we added a table describing the name of the miRNA family, the name of family members, the specific target of each family member, and the therapeutic strategies. We did not add the mechanism of action in the table since already reported in Figure 1.

  1. Lines 48-54 should be revised to cover miRNA biogenesis and mode of action to introduce the concept to those who might not be familiar with it.

We added a description of miRNAs biogenesis pathway and of their mechanism of action.

  1. There are just too many grammar and typing errors that require extensive proofreading by a native speaker. Some parts of the manuscript are rather cohesive and well-written whereas some other parts are not.

We extensively proofread the manuscript to correct the grammar and typing errors; we also rewrote some sentences to improve the readability of the manuscript.

  1. The authors should cite more current studies.

We thank the reviewer for the suggestion, we cited the most updated studies regarding the discussed miRNAs.

  1. There are inconsistencies in the use of terms. For example, the authors use “microRNA”, “miRNA” and “miR” interchangably. Similarty, “Author et al”, “Author et colleagues”, or “Author and colleagues”, giving the impression that different parts of the manuscript has been written by different authors without going over the last version. Please revise accordingly.

To improve the consistency in the use of terms throughout the manuscript, we applied the following modifications. Regarding the use of abbreviations of “microRNA”, we used “miRNA” when describing it in a general way (e.g. “miRNAs are small endogenous single-stranded noncoding RNA molecules”), while we used “miR” when describing a specific microRNA (e.g. miR-15a). Regarding how we spelled the citations, we used the form “Autor and colleagues”.

  1. I suggest “Role of miRNAs in ........” as the title.

We changed the title to “Role of microRNAs in Chronic Lymphocytic Leukemia”.

  1. There are several paragraphs consisting of single sentences. Please combine such paragrapsh with the preceeding or following paragraphs. Please avoid paragraphs of single sentences.

As suggested, we combined paragraphs consisting of single sentences with the preceding or following paragraphs.

We fixed all the following points:

  • Line 55, “microRNAs” should be “miRNA”
  • Line 64, “a 30 kb at” should be “a 30 kb region at”; also, “loss” should be “lost”
  • Line 65, “represent” should be “represents”
  • Line 81, “consists in” should be “consists of”—there are several such errors throughout the text.
  • Line 89, “abrogate” should be “abrogates”
  • Line 139, “does” should be “do”; lines 142-143, “downregulate” and transactivate” should be past tense.
  • Line 171, “find” should be “found”
  • Line 191, “over expression” à overexpression
  • Line 198, “egpressed” should be “expressed”
  • Line 203, “ifentifie” should be “identified”
  • Line 226, “which” should be “whose”
  • Line 228, “an own” should be “its own”
  • Line 235, “transcrib” is a bit awkward. Maybe “activate”?
  • Line 284 act à acts
  • Line 290 were à was
  • Line 299, results à result
  • Lines 305-307- This sentence is awkward
  • Line 320, 50-UTR à 5’UTR
  • Line 328, “Unmuted ...have” à Unmutated..... has... Please change all “unmuted” to “unmutated”
  • Lines 438-439, hamper, allow à hampers, allows
  • Line 481, orcheastrates à orchestrate; line 486, are à is; line 488, play à plays
  • “In vitro” and “in vivo” should be italicized.
  • Many more typing and grammar errors......

Reviewer 2 Report

The review “Role of miRNA in Chronic Lymphocytic Leukemia” provides a detailed description of a small subset of microRNAs that are abnormally expressed in CLL, detailing the causes of their abnormal expression, their role in the pathogenesis of CLL, and possible use in CLL therapy. The review is well written, well structured, but, unfortunately, one gets the feeling that the authors were in too much of a hurry to prepare it, so they did not find the opportunity to carefully reread it and correct minor shortcomings noted by the reviewer.

The title. It seems to me that it would be better to write “microRNAs” in the plural, because this review is not about one specific miRNA, but about several molecules.

Line 19. MicroRNA should be written with a small letter.

Line 21. You have already entered the abbreviation "miRNA" instead of "microRNA". Please use the abbreviation in the future or do not enter it at all.

Line 34. Missing space before links [2-4]. Further, there are the same blots.

Line 38. Tumor Protein 53 or TP53, the full name of the gene does not require an additional P in front of the digits, as it denotes a protein.

Line 63. Calin and colleagues or Calin et al. Further, it is desirable to use one option throughout the text.

Line 253. “MicroRNA” reduction to “miR” has not been previously introduced.

Line 270. “level was associated” or “levels were associated”. The first option is better.

Line 358. The abbreviation "oncomiR" is relatively well known, but still authors should not use it without first deciphering it.

Line 443. Dear authors, I also prefer the British spelling of the word “tumour”, but MDPI prefers the American one. Moreover, this is the only case you wrote "tumour".

When citing an article, attention should be paid to the correct common spelling "Author and colleagues" or "Author et al.". The spelling should be corrected throughout the manuscript.

Author Response

Dear Editor,

Thank you for giving us the opportunity to revise and improve our manuscript “Role of microRNAs in Chronic Lymphocytic Leukemia”. We also want to thank the reviewers for their constructive suggestions.

The main text has been edited to fix spelling errors and improve readability following the reviewers’ comments. The point-by-point replies to the major reviewers’ comments are below.

Changes in the manuscript are in red.

Reviewer #2

The title. It seems to me that it would be better to write “microRNAs” in the plural, because this review is not about one specific miRNA, but about several molecules.

We thank the reviewer for the suggestion, we changed the title of the Review to “Role of microRNAs in Chronic Lymphocytic Leukemia”.

When citing an article, attention should be paid to the correct common spelling "Author and colleagues" or "Author et al.". The spelling should be corrected throughout the manuscript.

We corrected throughout the manuscript the spelling of citations in “Author and colleagues”.

We fixed all the following points:

Line 19. MicroRNA should be written with a small letter.

Line 21. You have already entered the abbreviation “miRNA”; instead of “microRNA”. Please use the abbreviation in the future or do not enter it at all.

Line 34. Missing space before links [2-4]. Further, there are the same blots.

Line 38. Tumor Protein 53 or TP53, the full name of the gene does not require an additional P in front of the digits, as it denotes a protein.

Line 63. Calin and colleagues or Calin et al. Further, it is desirable to use one option throughout the text.

Line 253. “MicroRNA” reduction to “miR” has not been previously introduced.

Line 270. “level was associated” or “levels were associated”. The first option is better.

Line 358. The abbreviation “oncomiR”; is relatively well known, but still authors should not use it without first deciphering it.

Line 443. Dear authors, I also prefer the British spelling of the word “tumour”, but MDPI prefers the American one. Moreover, this is the only case you wrote “tumour”.

Round 2

Reviewer 1 Report

The authors have mostly addressed all the points raised. Just as a very minor point, I have detected two paragraphs consisting of single sentences (lines 205-206 and 352-354). The authors should integrate them into the previous or next paragragh without comprimising the cohesion of the manuscript. 

Fine.

Author Response

We thank once again Reviewer#1 for his constructive and detailed revision.

1) I have detected two paragraphs consisting of single sentences (lines 205-206 and 352-354). The authors should integrate them into the previous or next paragragh without comprimising the cohesion of the manuscript. 

As suggested, we combined the paragraphs indicated with the preceding or following paragraphs.
